# Nuclear envelope protein MAN1 regulates clock through *BMAL1*

**Shu-Ting Lin[1†], Luoying Zhang[1†], Xiaoyan Lin[1†‡], Linda Chen Zhang[1], Valentina Elizabeth Garcia[1], Chen-Wei Tsai[1], Louis Ptáček[1,2], Ying-Hui Fu[1]\***

[1]Department of Neurology, University of California, San Francisco, San Francisco, United States; [2]Howard Hughes Medical Institute, University of California, San Francisco, San Francisco, United States

**Abstract** Circadian clocks serve as internal pacemakers that influence many basic homeostatic processes; consequently, the expression and function of their components are tightly regulated by intricate networks of feedback loops that fine-tune circadian processes. Our knowledge of these components and pathways is far from exhaustive. In recent decades, the nuclear envelope has emerged as a global gene regulatory machine, although its role in circadian regulation has not been explored. We report that transcription of the core clock component *BMAL1* is positively modulated by the inner nuclear membrane protein MAN1, which directly binds the *BMAL1* promoter and enhances its transcription. Our results establish a novel connection between the nuclear periphery and circadian rhythmicity, therefore bridging two global regulatory systems that modulate all aspects of bodily functions.

**\*For correspondence:** ying-hui.fu@ucsf.edu

[†]These authors contributed equally to this work

**Present address:** [‡]Neurogenetics Branch, National Institute of Neurological Disorders and Stroke, Bethesda, United States

**Reviewing editor**: Leslie C Griffith, Brandeis University, United States

## Introduction

Most organisms, ranging from cyanobacteria to humans, are governed by their circadian rhythms: endogenous and self-sustained oscillations with a period of roughly 24 hr, which manifest in diverse metabolic, physiological, and behavioral processes (*Ueda et al., 2005*). This internal pacemaker is charged with two important roles: to perpetuate its own rhythms and to regulate the expression of genes that are under circadian control. In mammals, this internal pacemaker consists of a complex network of transcriptional regulations, at the core of which is transcription activators BMAL1 (also known as ARNTL1 or MOP3) and CLOCK, which form heterodimers and regulate gene expression. Up to 15% of the organism's genome is regulated in a circadian manner (*Panda et al., 2002*; *Emery and Reppert, 2004*; *Zhang and Kay, 2010*). Well-studied examples include the transcription repressors (PERIODs and CRYPTOCRHOMEs) that bind to CLOCK/BMAL1 and suppress their own transcription, thereby forming a feedback loop. Since the identification and cloning of the first mammalian clock gene, *CLOCK*, two decades ago (*Vitaterna et al., 1994*), the field of chronobiology has uncovered many additional players that regulate circadian rhythms on a transcriptional and/or post-transcriptional level, and many more such candidates are currently being evaluated.

Recently, mutations in nuclear envelope (NE) proteins have been shown to cause a surprisingly broad range of inherited diseases, thus shedding light on roles played by the NE in global regulations at cellular and organismal levels (*Padiath et al., 2006*; *Dauer and Worman, 2009*). These diseases (often referred to as nuclear envelopathies or laminopathies) can impact muscle, nerve, fat metabolism, bone formation, and others. NE consists of outer and inner nuclear membranes (connected by nuclear pore complexes) and nuclear lamina. The inner nuclear membrane proteins (such as MAN1, LBR, LAP2, etc) include approximately 60 putative transmembrane proteins specifically retained in the inner nuclear membrane and most of them are poorly characterized (*Schirmer et al., 2005*). In metazoan, nuclear lamina is a protein mesh-like structure composed of type V intermediate filament proteins lamins

**eLife digest** If rodents, or indeed humans, are kept in constant darkness for a number of days, they continue to show patterns of sleep and wakefulness that repeat roughly every 24 hr. This internal 'circadian rhythm' controls many aspects of animal physiology, including body temperature, blood pressure, and hormone levels. It does so by regulating the expression of key genes: this means that the activity of the proteins encoded by these genes also varies in accordance with the circadian rhythm.

A second mechanism used by the body to coordinate gene expression on a large scale entails making adjustments to the membrane that surrounds the cell nucleus. This 'nuclear envelope' consists mainly of lipids, but it also contains proteins that bind DNA. These proteins regulate gene expression by controlling how easy it is for other proteins that activate or repress genes to gain access to specific DNA sequences.

Lin et al. now reveal that these mechanisms work together. The first evidence for this was the discovery that the levels of three specific nuclear envelope proteins influence, and are influenced by, circadian rhythms. In particular, two of these proteins control the activity of the third, which is known as MAN1. This protein in turn triggers the expression of a gene called *BMAL1*, which is one of the small number of 'clock genes' that are responsible for generating the internal circadian rhythm.

As well as adding to our knowledge of circadian biology and the nuclear envelope, this study reveals a mechanism by which cells can orchestrate the expression of large numbers of genes. Such mechanisms allow a wide range of physiological and behavioral processes to be co-ordinated.

(including A, C, B1, B2, and B3 types) and sits primarily underneath the inner nuclear membrane (*Zwerger and Medalia, 2013*). The idea of nuclear envelope components as transcription regulators in mammals is relatively new, conceived from the observation that gene-rich chromosomes are generally located in more internal nuclear regions, whereas gene-poor chromosomes are relegated to the periphery (*Spector, 2003*). Many NE components such as inner nuclear membrane proteins, nuclear lamina, and the nuclear pore complex, harbor DNA-binding domains that are involved in anchoring chromatin to the periphery (*Ulbert et al., 2006*; *Mekhail and Moazed, 2010*). Functional relevance of these positional distinctions became apparent as studies with yeast and flies revealed that the NE can sequester factors that affect gene transcription in both repressive and, surprisingly, activating manners (*Akhtar and Gasser, 2007*). Although recent findings highlight the important functions of the nuclear periphery, its relationship with the circadian clock has not been probed. Given the increasing awareness of the global roles that these two systems play in myriad pathways, we set out to investigate the possibility that these seemingly separate pathways are connected and can work synergistically in regulating diverse functions.

## Results

### Nuclear envelope participates in circadian regulation

In order to investigate whether NE proteins are involved in circadian regulation, we began by focusing on lamin B1 since it has been shown to play a role in transcriptional regulation (*Hutchison, 2002*; *Shevelyov et al., 2009*). In vivo oscillation of lamin B1 (*Lmnb1*) expression patterns (both RNA and protein levels) were confirmed using mouse tissues from suprachiasmatic nuclei (SCN), kidney, and liver (*Figure 1A,B*). To test whether the level of lamin B1 affects the molecular clock, we examined the protein expression patterns for the core clock gene PERIOD2 (PER2) using *Lmnb1* heterozygous knock out (homozygosity is lethal) and *LMNB1* wild-type BAC transgenic mice (*Vergnes et al., 2004*; *Heng et al., 2013*). Oscillating PER2 expression patterns were phase delayed in *Lmnb1* heterozygous knock out mice and phase advanced in *LMNB1* BAC transgenic mice (overexpression) when compared to wild-type control mice (*Figure 1C*), suggesting that the level of lamin B1 can modulate circadian clock. However, neither *Lmnb1* heterozygous knock out mice nor *LMNB1* BAC transgenic mice demonstrated significant output behavioral change (*Figure 1—figure supplement 1*). To expand the investigation, we chose to include two additional NE proteins that are known to associate with lamin

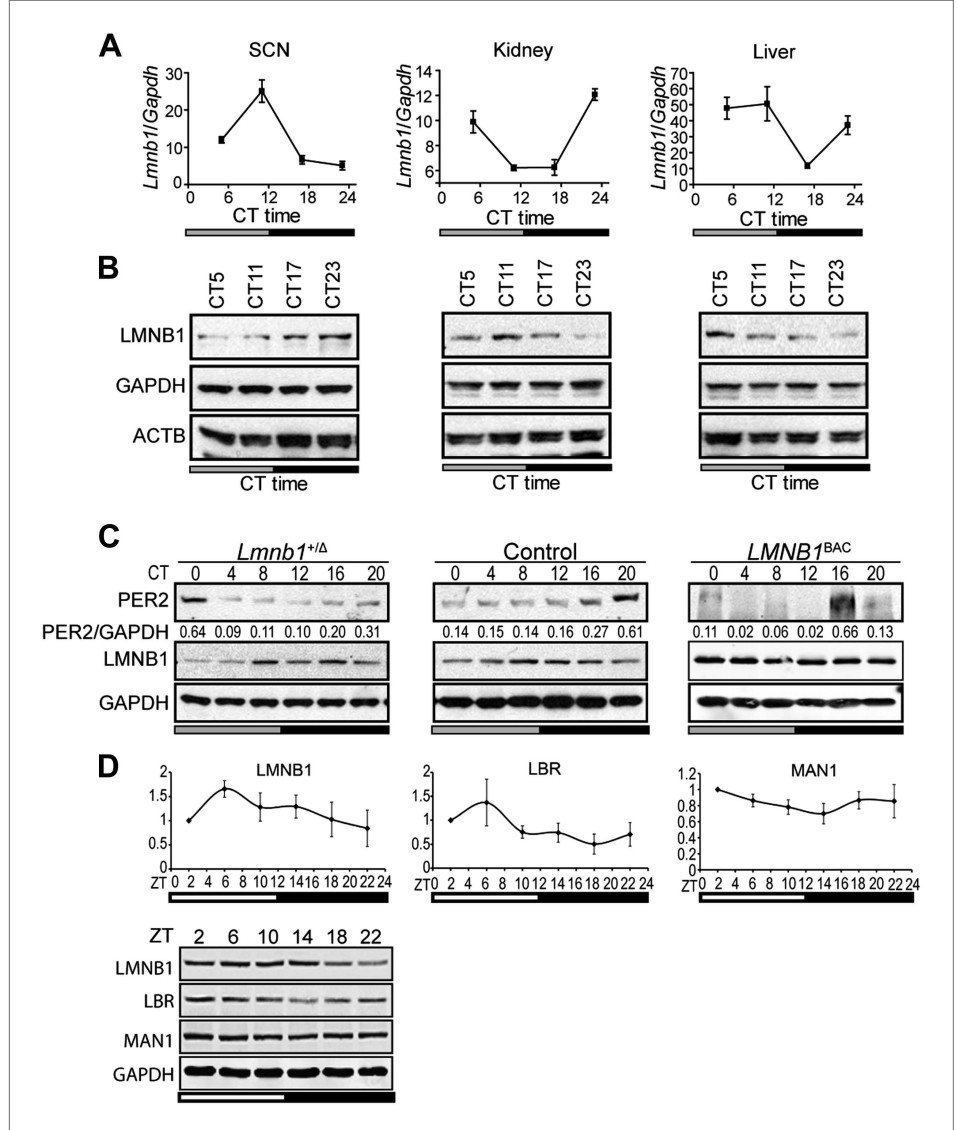

**Figure 1**. Lamin b1 regulates the circadian clock. Expression levels of lamin b1 from SCN, kidney, and liver extracts in C57BL/6J mice (**A** and **B**). (**A**) mRNA levels of *Lmnb1* and *Gapdh* were assayed at indicated circadian times (CT, n = 4). (**B**) Representative immunoblots show the levels of LMNB1, GAPDH, and β-ACTIN. (**C**) Representative immunoblots show PER2 (with intensity values indicated at the bottom) and LMNB1 abundance in *Lmnb1+/Δ*, wild-type and *LMNB1*BAC liver extracts. (**D**) Expression patterns of LMNB1, LBR, and MAN1 in C57BL/6 mouse livers at indicated Zeitgeber times (ZT) (n = 3). Quantifications (top panels) of Western blots (bottom panel) were obtained by using GAPDH as a loading control. Data represent means ± SD.

The following figure supplement is available for figure 1:

**Figure supplement 1**. Mice with altered LMNB1 levels do not exhibit altered behavioral rhythms.

B1, LBR, and MAN1. We found that LBR and MAN1 expression also oscillate, albeit mildly for MAN1 (*Figure 1D*).

To determine if these NE genes passively receive cues from the core clock apparatus or if their protein products also actively play a role in maintaining circadian rhythms, we altered their protein levels in human osteosarcoma U2OS cells that express a luciferase reporter gene under the control of mouse *Bmal1* promoter (*Bmal1*-Luc) and examined circadian period in cell culture (*Vollmers et al., 2008*). siRNA-induced reduction of *LMNB1*, *LBR*, and *MAN1* in this cell-based system resulted in a longer circadian period (τ) (*Figure 2A*), whereas the over-expression of all three led to a shorter τ (*Figure 2B*). Cells transfected with

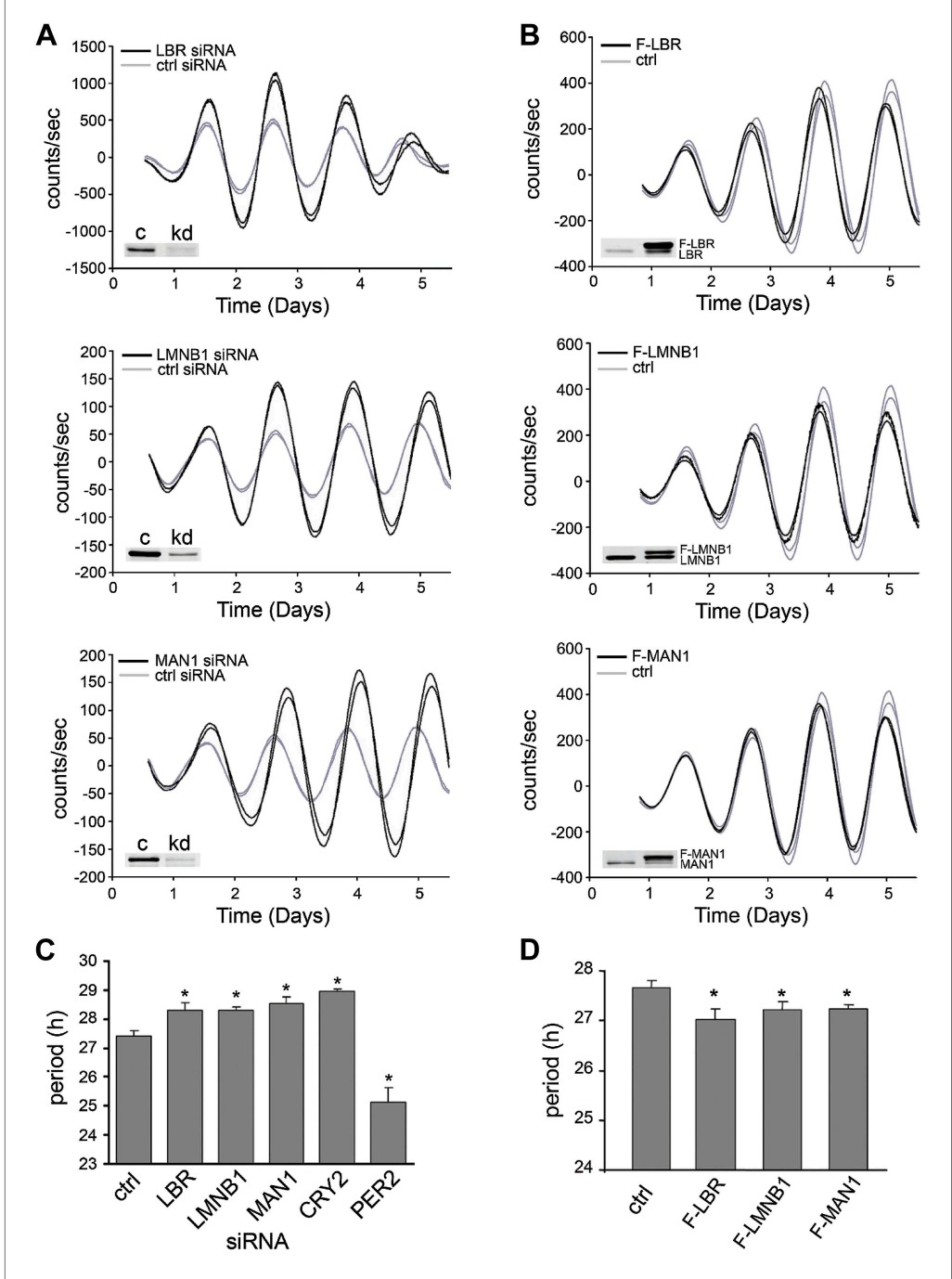

**Figure 2**. LBR, LMNB1, and MAN1 are necessary for normal circadian rhythms. Two representative traces of real-time bioluminescence analyses are shown for each, and Western blot verification of down-regulation or over-expression is demonstrated in the inset images. (**A**) Period was lengthened when *LBR, LMNB1*, or *MAN1* was knocked down. (**B**) Over-expression of FLAG-tagged LBR (F-LBR), LMNB1 (F-LMNB1), or MAN1 (F-MAN1) shortened period compared to cells transfected with empty vector (ctrl). (**C** and **D**) Summary of period in (**A** and **B**). *CRYPTOCRHOME2* (*CRY2*) and *PER2* siRNA knockdowns served as positive controls. Data represent means ± SD.

The following figure supplements are available for figure 2:

**Figure supplement 1**. Over-expressing or knocking down nuclear envelope components alters circadian rhythms in flies.

**Figure supplement 2**. The mRNA levels of nuclear envelope genes are reduced in the corresponding knockdown flies.

*LBR*, *LMNB1*, and *MAN1* siRNA lengthened τ by 54–69 min (n ≥ 4, *p < 0.05), when compared with control siRNA (τ = 27.39 ± 0.22 hr, n = 8) (**Figure 2C**). On the other hand, overexpression of FLAG-tagged LBR, LMNB1, or MAN1 shortened τ by 25.8–37.8 min (n ≥ 4, *p < 0.05) compared to empty vector controls (τ = 27.7 ± 0.15, **Figure 2D**). These changes in τ together with the altered phase of *Lmnb1* heterozy-gous knock out mice and over-expressing mice suggest that these NE proteins participate in modulating circadian clock and therefore could impose significant impacts on downstream biological pathways.

The effects of NE proteins on *Drosophila* circadian clock were also examined. Consistent with the mammalian data, over-expressing d*Lamin* (d*Lam*) (**Padiath et al., 2006**) in circadian neurons with *cryptochrome* (*cry*)GAL4-39 and *cry*GAL4-16 (**Emery et al., 2000**) in vivo resulted in substantially shortened periods of behavioral rhythms in constant darkness compared to GAL4 controls (but not to UAS*LMNB1*/+, **Table 1**; **Figure 2—figure supplement 1A**). Knocking down d*Lam* in circadian neurons lengthened the period (**Table 2**; **Figure 2—figure supplement 1B**). On the other hand, over-expressing d*MAN1* and d*LBR* lengthened the period (**Table 1**; **Figure 2—figure supplement 1A**), while knocking down d*MAN1* also lengthened period (**Table 2**; **Figure 2—figure supplement 1B**). Besides altering the period, most of these manipulations reduced the amplitude of behavioral rhythms as indicated by the reduced power values. In addition, we have assessed the mRNA levels of d*MAN1*, d*Lam*, and d*LBR* to confirm knockdown (**Figure 2—figure supplement 2**). Taken together, these results indicate that NE proteins also participate in the regulation of fly clock.

## Lamin B1 and LBR likely act through MAN1

We next explored the relationship of *LBR*, *LMNB1*, and *MAN1* by examining mRNA and protein levels while knocking them down one at a time. Both *LBR* and *LMNB1* knockdown significantly decreased the transcript level of *MAN1* (by 15% and 40%, respectively) (**Figure 3A**). The effects of *LBR* or *LMNB1* knockdown on MAN1 expression are even more dramatic at the protein level, with 54% and 44% reductions, respectively (**Figure 3B**). Moreover, knockdown of *LBR* expression reduces the amount of LMNB1 protein by 32%, which is consistent with the observation that reduction of *LBR* expression in the fibroblasts of patients harboring a heterozygous *LBR* mutation results in the abolition of LMNB1 protein (**Gaudy-Marqueste et al., 2010**), whereas a decrease in *LMNB1* does not significantly affect *LBR* expression. *MAN1* knockdown also does not change the expression of *LBR* and *LMNB1*, either at the mRNA or protein level (**Figure 3**). These results suggest that *MAN1* is modulated by LBR and LMNB1, and thus the effects of LBR and LMNB1 on the clock are at least partially through MAN1. Therefore, we further investigated the effects of MAN1 on the molecular clock.

## MAN1 regulates the clock by promoting *BMAL1* expression

A lengthened period due to decreased MAN1 may arise from altered regulation of core clock proteins and/or altered transcription of core clock genes. Either of these effects would result in disruptions of the

**Table 1.** Over-expressing NE genes alters the behavioral period in flies

| Genotype | Period (hr) | Power | Rhythmic% | N |
|---|---|---|---|---|
| UAS*MAN1*/+ | 23.8 ± 0.0 | 90 ± 5 | 96 | 76 |
| *cry*GAL4-39/UAS*MAN1* | 26.4 ± 0.3*,† | 31 ± 6*,† | 64 | 28 |
| UAS*MAN1*/+; *cry*GAL4-16/+ | 26.8 ± 0.2*,† | 40 ± 7*,† | 66 | 32 |
| UAS*Lam*/+ | 23.9 ± 0.0 | 87 ± 6 | 93 | 55 |
| *cry*GAL4-39/UAS*Lam* | 23.8 ± 0.2† | 33 ± 7*,† | 73 | 22 |
| UAS*LMNB1*/+; *cry*GAL4-16/+ | 24.4 ± 0.2† | 5 ± 2*,† | 26 | 31 |
| UAS*LBR*/+ | 23.7 ± 0.0 | 59 ± 5 | 79 | 67 |
| *cry*GAL4-39/UAS*LBR* | 25.6 ± 0.1*,† | 44 ± 6† | 79 | 33 |
| UAS*LBR*/+; *cry*GAL4-16/+ | N/A | 1 ± 0*,† | 0 | 32 |
| *cry*GAL4-39/+ | 24.8 ± 0.1 | 86 ± 5 | 93 | 57 |
| *cry*GAL4-16/+ | 25.6 ± 0.1 | 86 ± 5 | 94 | 64 |

*One-way ANOVA compared to UAS control lines, p < 0.001.
†One-way ANOVA compared to GAL4 control lines, p < 0.001.

**Table 2.** Knocking-down NE genes lengthens the behavioral period in flies

| Genotype | Period (hr) | Power | %Rhythmic | N |
|---|---|---|---|---|
| UAS*MAN1RNAi* | 24.4 ± 0.1 | 110 ± 10 | 100 | 14 |
| UAS*MAN1RNAi*;*cry*GAL4-39/+; UAS*dcr2*/+ | 26.3 ± 0.3*,† | 29 ± 6*,† | 80 | 15 |
| UAS*MAN1RNAi*; UAS*dcr2*/+;*cry*GAL4-16/+ | 27.9 ± 0.4*,† | 39 ± 8*,† | 69 | 16 |
| UAS*LamRNAi*/+ | 23.6 ± 0.1 | 129 ± 12 | 100 | 16 |
| *cry*GAL4-39/+; UAS*LamRNAi*/UAS*dcr2* | 25.1 ± 0.4* | 26 ± 17*,† | 50 | 14 |
| UAS*LamRNAi*/UAS*dcr2*; *cry*GAL4-16/+ | 26.8 ± 0.1*,† | 126 ± 17 | 100 | 13 |
| UAS*LBRRNAi*/+ | 23.5 ± 0.0 | 112 ± 14 | 100 | 15 |
| *cry*GAL4-39/UAS*LBRRNAi*; UAS*dcr2*/+ | 24.9 ± 0.1 | 61 ± 12* | 80 | 15 |
| UAS*LBRRNAi*/UAS*dcr2*; *cry*GAL4-16/+ | 24.6 ± 1.7 | 12 ± 4*,† | 44 | 16 |
| *cry*GAL4-39/+; UAS*dcr2*/+ | 24.9 ± 0.1 | 69 ± 12 | 88 | 16 |
| UAS*dcr2*/+;*cry*GAL4-16/+ | 26.1 ± 0.1 | 87 ± 10 | 94 | 16 |

*One-way ANOVA compared to UAS*RNAi* control lines, p < 0.05.
†One-way ANOVA compared to control lines with GAL4 and UAS*dcr2*, p < 0.05.
*dicer2* (*dcr2*) is co-expressed to enhance the effects of RNAi.

stoichiometry and temporal control of the dynamics of the core circadian feedback loops. Given what is known regarding the role of NE proteins in transcriptional regulation, we tested whether reductions in MAN1 expression would affect the transcription of clock genes. We examined the circadian oscillation of core clock genes at the mRNA level and found that only *BMAL1* showed a clear difference wherein overall mRNA levels were down-regulated to half the levels of controls (***Figure 4***). Western blots also showed lower expression of BMAL1 when MAN1 was knocked down (***Figure 4—figure supplement 1***). The non-oscillatory *CLOCK* showed no significant change of either transcript or protein levels (***Figure 4***, ***Figure 4—figure supplement 1***). The conserved mRNA levels of *REV-ERBα and RORα* in MAN1 knock-down cells suggest that the reduced *BMAL1* expression is not caused by altered transcriptional activation of *REV-ERBα*, a *BMAL1* repressor, or transcriptional repression of *RORα*, a *BMAL1* activator.

To confirm MAN1 regulates the clock by targeting *BMAL1*, we over-expressed *Bmal1* in *MAN1* knockdown U2OS *Bmal1*-Luc cells. Knocking down *MAN1* lengthened the period in control cells as described above (***Figure 2***), whereas cells over-expressing sufficient *Bmal1* did not demonstrate period lengthening compared to cells without *MAN1* knockdown (***Figure 4—figure supplement 2***), suggesting that the lengthened period caused by *MAN1* deficiency is due to reduction of BMAL1.

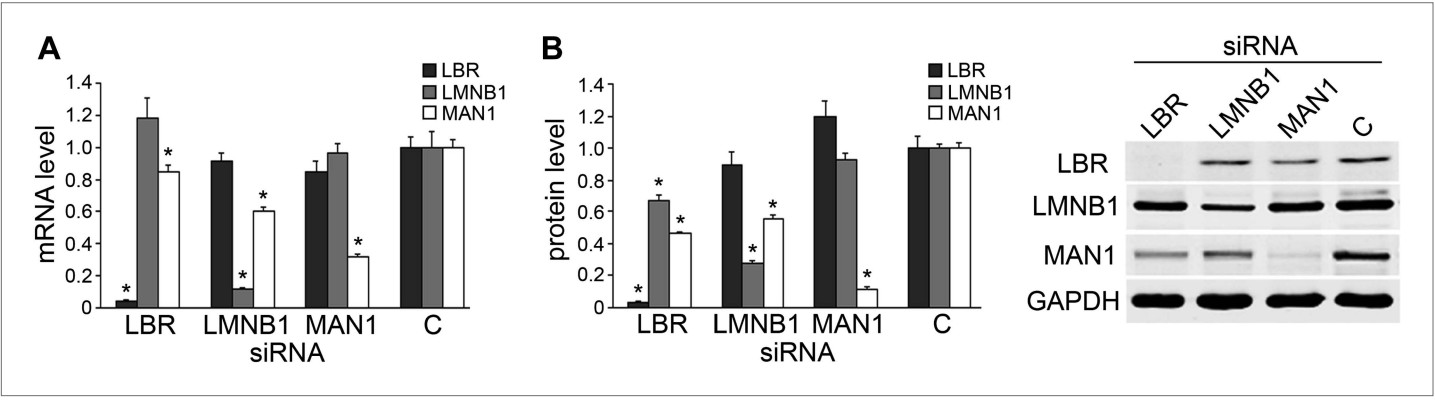

**Figure 3**. Knocking down *LBR*/*LMNB1* reduces *MAN1* mRNA and protein levels but not vice versa. Assessing mRNA (**A**) and protein (**B**) levels of *LBR*, *LMNB1*, and *MAN1* while knocking them down one at a time in U2OS cells via RNAi. (**A**) mRNA levels of *LBR*, *LMNB1*, and *MAN1* in each of the three knockdown conditions were quantified using qRT-PCR (n = 14, *p < 0.05). (**B**) MAN1 was significantly down-regulated when *LBR* or *LMNB1* was knocked down (n = 14 *p < 0.001). The error bars represent SEM (left panel). Representative immunoblots show the protein levels of LBR, LMNB1 and MAN1 (right panel).

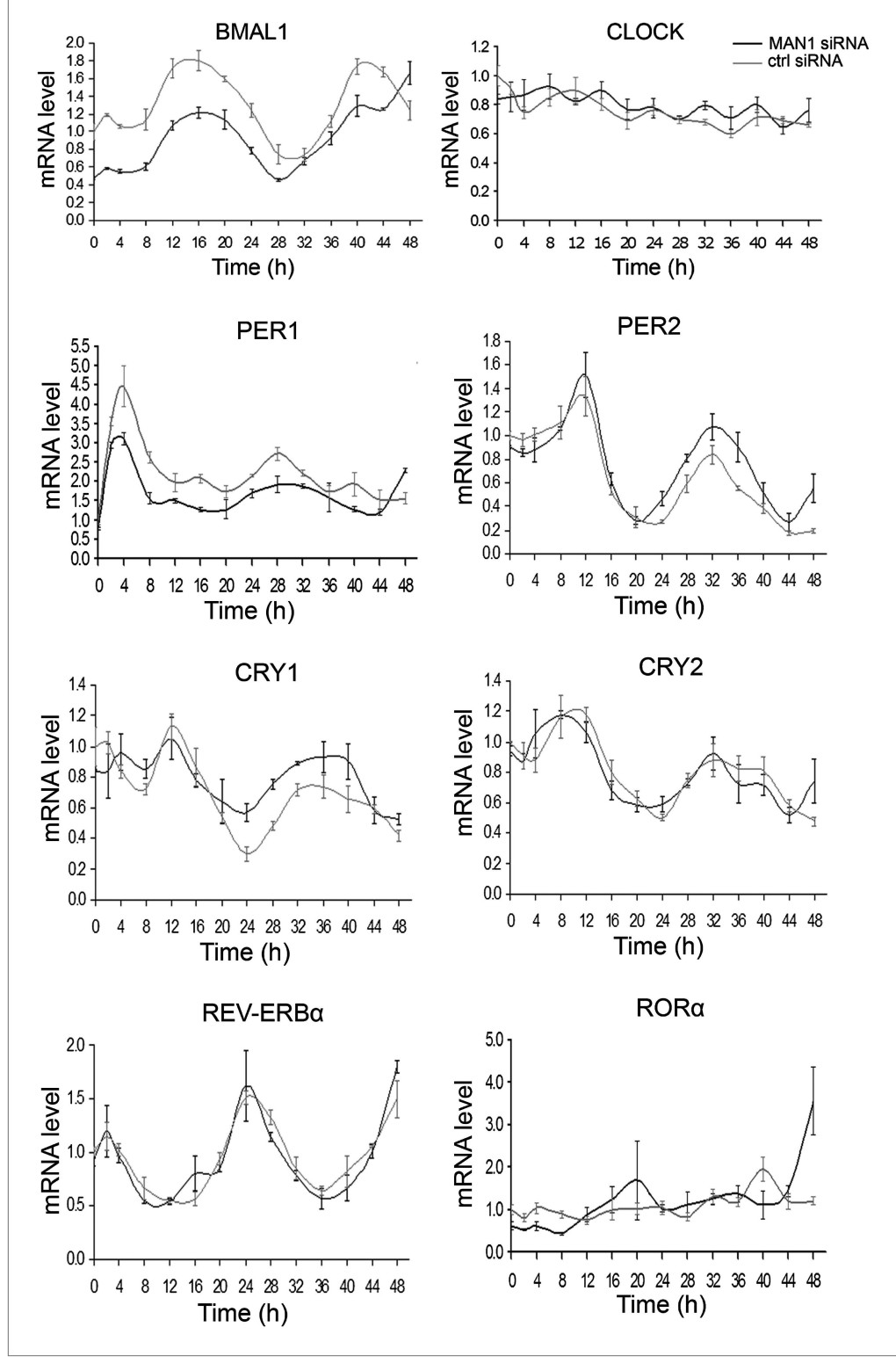

**Figure 4**. Knocking down *MAN1* reduces the levels of *BMAL1* mRNA. Each graph shows cells transfected with *MAN1* siRNA vs ctrl siRNA. Time 0 indicates the moment that U2OS cells were treated with dexamethasone (100 nM). Data are presented as means ± SEM, n = 3.

*Figure 4. Continued on next page*

*Figure 4. Continued*

The following figure supplements are available for figure 4:

**Figure supplement 1**. Knocking down *MAN1* reduces BMAL1 protein levels.

**Figure supplement 2**. Over-expressing *Bmal1* suppresses the period lengthening effect of *MAN1* knockdown.

Together, these results indicate that MAN1 functions to promote *BMAL1* expression, and thus exerting effects on the clock.

Consistent with the cell culture data, over-expressing MAN1 in all clock cells in flies using a *timeless* (*tim*) GAL4 driver (*Emery et al., 1998*) resulted in a significantly increased level of *cycle* (*cyc*) mRNA, the *Drosophila BMAL1* homologue (*Rutila et al., 1998*; *Figure 5A*). The mRNA level of core clock gene *tim* was also significantly elevated. In addition, we have assessed the levels of *MAN1* mRNA to confirm over-expression (*Figure 5B*). We also examined the effect of knocking down *MAN1* in clock cells but did not observe altered *cyc* mRNA levels (*Figure 5—figure supplement 1*).

## MAN1 enhances *BMAL1* transcription

A luciferase reporter assay using HEK293 cells was used to further investigate the effect of MAN1 on *BMAL1* transcription. *MAN1* knockdown decreased *Bmal1*-Luc activity by 72% (*Figure 6A*), whereas overexpression of FLAG-MAN1 increased the luciferase activity by more than twofold vs cells transfected with empty vector (*Figure 6B*). Similar results were obtained with a longer, human *BMAL1* promoter (*Figure 6C*). These data indicate that MAN1 may play a role in circadian regulation by activating the promoter of *BMAL1*.

Previously, MAN1 has been shown to exert antagonistic regulatory functions on signal transduction through its binding to R-SMADs (*Osada et al., 2003*; *Raju et al., 2003*; *Hellemans et al., 2004*; *Lin et al., 2005*; *Pan et al., 2005*; *Cohen et al., 2007*) and two types of R-SMADs are found in mammals: TGFβ-responsive (SMAD2 and SMAD3) and BMP-responsive (SMAD1, SMAD5, and SMAD8). To determine whether R-SMADs have an effect on *BMAL1* transcription, we first expressed R-SMADs individually in HEK293 cells transfected with *BMAL1*-Luc to determine which R-SMAD/s is/are involved in regulating *BMAL1* transcription. Expressing SMAD1, SMAD5, SMAD8, and SMAD3 had no significant effect on *BMAL1* transcription but SMAD2 showed significant enhancing effect, suggesting a possible regulatory function by SMAD2 in *BMAL1* regulation (*Figure 7A*). The enhancing action of SMAD2 was then examined together with MAN1 to determine whether there is interplay between MAN1 and SMAD2 on *BMAL1* promoter activity. Intriguingly, MAN1 further augmented the enhancing effect of SMAD2 on *BMAL1* in an additive manner, indicating that the positive regulatory function of MAN1 and SMAD2 on *BMAL1* might be independent of each other (*Figure 7B*).

Since *BMAL1* transcription is regulated by RORα and REV-ERBα, we wondered whether the effect of MAN1 on *BMAL1* is influenced by RORα/REV-ERBα. MAN1 increases *BMAL1* transcription in the HEK293 luciferase reporter assay but this effect was overshadowed by the presence of either RORα or REV-ERBα, and the impact of RORα and REV-ERBα on *BMAL1* was not influenced by the presence of MAN1 (*Figure 7C*). In addition, the effect of MAN1 was also not significantly altered by mutating the RORE sequence (+3 ~ +13 and +39 ~ +48) in the *BMAL1* promoter (*Figure 7D*), which serves as the DNA binding target of RORα and REV-ERBα. Together, these data suggest that the effect of MAN1 on *BMAL1* transcription is not through the RORE and does not require or impact RORα and REV-ERBα.

## MAN1 binds to the *BMAL1* promoter

Since MAN1 does not execute its function through RORE, we investigated the promoter region of *BMAL1* to determine what is necessary for the enhancing effect of MAN1. A series of deletion constructs of the *BMAL1* promoter were generated for luciferase assays and a 900 bp region (−795 ~ +106) was identified to be the region harboring the necessary DNA sequence for the regulatory effect of MAN1 on *BMAL1* (*Figure 8A*, *Figure 8—figure supplement 1A*).

MAN1 has an N-terminal L̲AP2-E̲merin-M̲AN1 (LEM) domain, two transmembrane segments in the middle, a unique DNA binding domain, and a C-terminal RNA recognition motif (RRM) that is required for its binding with R-SMADs (*Caputo et al., 2006*). We next examined the domain of MAN1 necessary for its effect on *BMAL1*. Two truncation constructs of MAN1 were generated, one without the DNA binding domain (amino acids 707–725) and the other without the RRM domain (amino acids

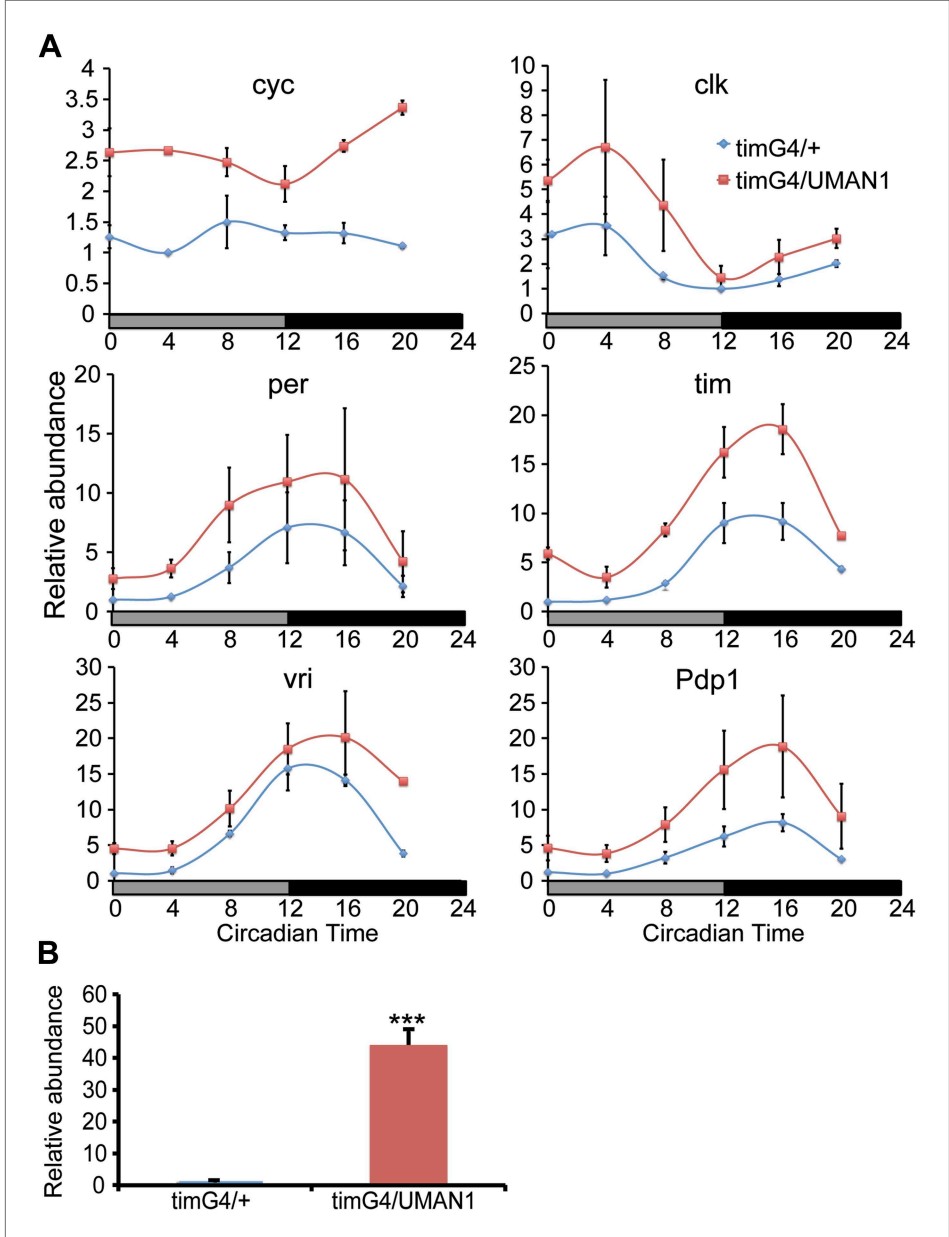

**Figure 5**. MAN1 increases *cyc* mRNA levels. (**A**) Plots of relative mRNA abundance vs CT for clock genes from whole head extracts of *tim*GAL4/+ and *tim*GAL4/UAS*MAN1* flies during the first day of DD determined by qRT-PCR (n = 2). Gray box indicates subjective light period and black box indicates dark period. Significant effect of genotypes (Two-way ANOVA) were found for *cyc* (p = 0.0278) and *tim* (p = 0.0161). For each time series, the value of the lowest time point was set to 1. (**B**) Plot of relative mRNA abundance for *MAN1* from whole head extracts of *tim*GAL4/+ and *tim*GAL4/UAS*MAN1* flies determined by qRT-PCR (n = 6, Student's *t* test, ***p < 0.001). The value of *tim*GAL4/+ in one experiment was set to 1. Error bars represent SEM.

The following figure supplement is available for figure 5:

**Figure supplement 1**. *cyc* transcript level is not altered in *MAN1* knock-down flies.

---

760–911) (*Pan et al., 2005*; *Figure 8—figure supplement 1B*). In addition, we utilized a substitution mutant of MAN1 (YV-DD) (*Pan et al., 2005*), containing two amino acid alterations in the RRM that nullify the ability of MAN1 to antagonize R-SMADs. These constructs were then used to test their transcriptional enhancing effect on *BMAL1* promoters (either full length 3.4 kb or 2.4 kb [−2300 ~

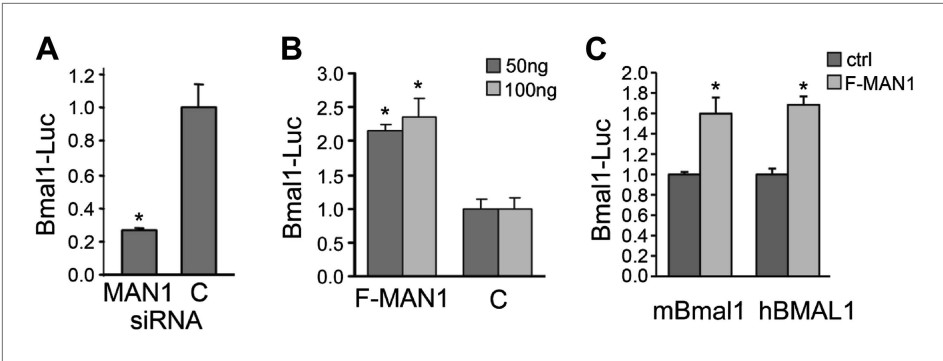

**Figure 6**. MAN1 promotes *BMAL1* transcriptional activity. (**A**) Reduction of *MAN1* transcripts (13 nM siRNA) reduced *Bmal1* promoter activity (n = 3, *p < 0.001). (**B**) Over-expression of FLAG-tagged MAN1 (F-MAN1) enhanced *Bmal1*-Luc activity (n = 3, *p < 0.001). (**C**) Over-expression of FLAG-tagged MAN1 (F-MAN1) enhanced luciferase activities driven by m*Bmal1* promoter (530 bp) or h*BMAL1* promoter (3.4 kb) (n = 3, *p < 0.05). Error bars represent SD.

+105]). Intriguingly, the RRM truncation mutation lost the enhancing effect on the *BMAL1* promoter (*Figure 8B*). The DNA binding domain truncation mutation also lost the activating effect on *BMAL1*, and this effect can be produced by simply mutating three positively charged amino acids (RKK) at amino acids 709–711 (*Figure 8C*, *Figure 8—figure supplement 1C*). These results imply that the effect of MAN1 on *BMAL1* transcription requires potential DNA binding ability of MAN1 as well as interaction with a protein partner (possibly SMAD2) through RRM.

Since the DNA binding domain is required for the effect of MAN1 on the *BMAL1* promoter, we next tested whether MAN1 directly binds to the *BMAL1* promoter. Chromatin immunoprecipitation (ChIP) analysis revealed that the region −237 bp to +45 bp from transcriptional start site was pulled down by MAN1 indicating a direct interaction (*Figure 8D*). All together, these data suggest that MAN1 binds to the *BMAL1* promoter region (−237 bp to +45 bp) to enhance its transcription.

## Discussion

The nuclear envelope plays essential roles in diverse cellular functions including global regulation of gene expression. Interestingly, another global regulatory mechanism is the molecular clock that modulates our body and cellular daily rhythm. We sought to see whether there is cross talk between these two regulatory mechanisms. Our studies revealed that some components of nuclear envelope do show daily oscillations, indicating that nuclear envelope is subject to clock control. On the other hand, we found that one of the major transcription activators of the molecular clock, *BMAL1*, is regulated by one of the nuclear envelope proteins, MAN1. Thus, there is reciprocity between these two global regulatory mechanisms.

The nuclear envelope physically separates genomic DNA from the cytoplasm and functions as a signaling control center. An increasing number of human diseases are recognized to be caused by mutations in genes encoding nuclear envelope proteins and hence, termed 'envelopathies' (*Dauer and Worman, 2009*). Several inner nuclear membrane proteins are known to regulate critical signaling pathways and act as intranuclear regulators of signaling pathways that receive and transduce signals from extracellular cues. The nuclear lamina provides structural support for the nucleus and the nuclear envelope; however, lamins and their associated proteins are also involved in most of the nuclear processes. Lamin B1 is essential for brain development and is required for the integrity of the nuclear lamina (*Padiath et al., 2006*; *Coffinier et al., 2011*). Interestingly, no abnormality has been reported for the heterozygous *Lmnb1* knock out mice (*Stewart et al., 2007*; *Coffinier et al., 2011*), and it is estimated that these heterozygous mice still express 70% of lamin b1 (SG Young, personal communication). Consistent with these reports, we did not find significant phenotype including circadian behavior change in these mice despite the clear phase shift on PER2 oscillation in the liver. It is possible that the oscillation of one or more additional core clock components are also altered by reduced lamin b1 level and these additional alterations can compensate the effect of PER2 phase shift on output behavior. Alternatively, it is also possible that the SCN clock is resilient to perturbations caused by reduced lamin b1 level, hence we can only observe alterations in peripheral clocks. Further investigation is necessary to reveal the mechanism leading to the findings we report here.

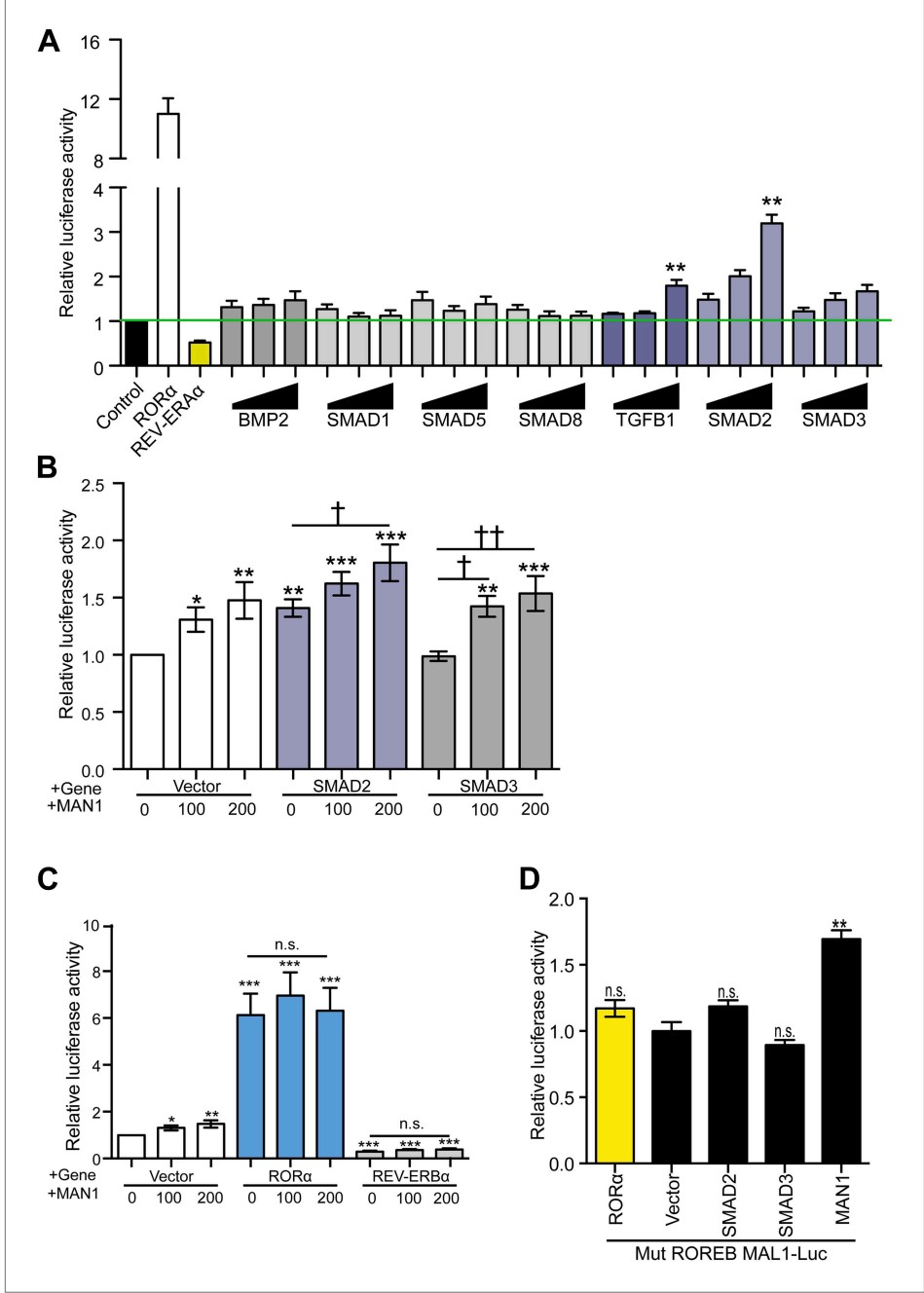

**Figure 7**. MAN1 and SMAD2 enhance *BMAL1* transcription. (**A–D**) Luciferase reporter activities in transfected HEK293 cells. Cells transfected with indicated constructs in the presence of the 3.4 kb h*BMAL1* promoter for 48 hr and relative luciferase activities were measured in extracts and normalized to *Renilla* luciferase activities. Relative luciferase activities were shown on the y-axis. Values are means ± SEM, n = 3, \*\*p < 0.01 compared to control. †p < 0.05, ††p < 0.01 compared to MAN1 0, one-way ANOVA with Newman–Keuls test.

MAN1 belongs to the inner nuclear membrane LEM protein family (*Worman, 2006*; *Bengtsson, 2007*). LEM domains mediate the interaction with a chromatin-binding protein, barrier-to-autointegration factor (BAF), which has the ability to bind dsDNA, chromatin, histones, lamin binding proteins, and various transcription factors (*Liu et al., 2003*; *Worman, 2006*; *Bengtsson, 2007*). Therefore, LEM proteins have roles in gene regulation, chromatin organization, regulation of transcription factor activity at the nuclear periphery, and regulation of specific signal pathways. Both amino and carboxyl termini of MAN1

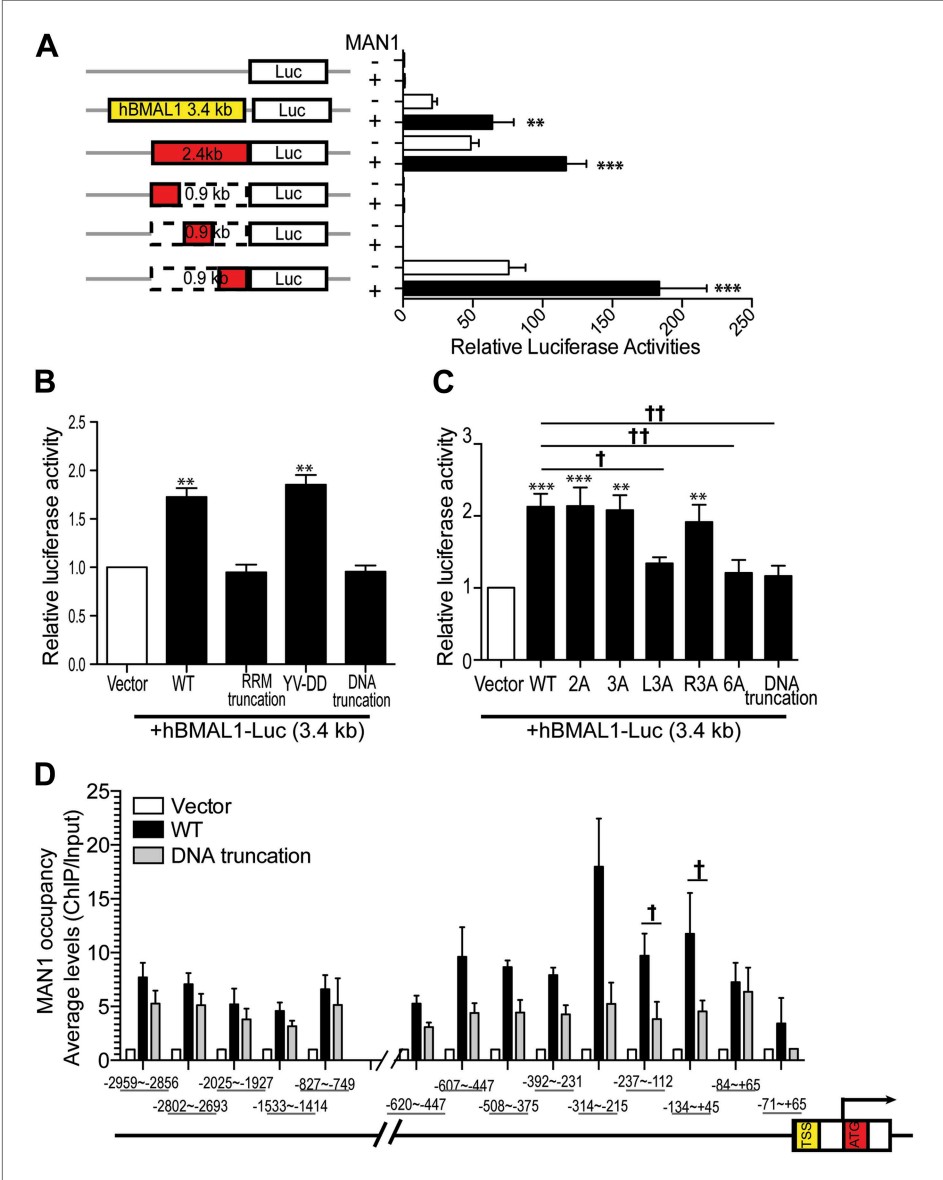

**Figure 8**. MAN1 binds to the *BMAL1* promoter to enhance its transcription. (**A**) Luciferase activities of deleted h*BMAL1*-promoter constructs in the absence or presence of MAN1 expression vectors. n = 3, Student's *t* test, **p < 0.01, ***p < 0.001. (**B** and **C**) Luciferase activities of the 3.4 Kb h*BMAL1-Luc* in the presence of MAN1 constructs as indicated. n = 3, **p < 0.01, ***p < 0.001 compared to control; †p < 0.05; ††p < 0.001 compared to WT MAN1. (**D**) ChIP analysis of MAN1 (WT or DNA binding truncation) for 14 segments of h*BMAL1* promoter region. Data represent pull-down relative to input. n = 6, †p < 0.05, compared to WT MAN1. One-way ANOVA with Newman–Keuls test. All data are presented as ratio of means ± SEM.

The following figure supplement is available for figure 8:

**Figure supplement 1**. Domain-specific interactions between MAN1 and the h*BMAL1* promoter.

---

are nucleoplasmic domains. The amino-terminal nucleoplasmic region of MAN1 (including LEM) binds to the nuclear lamins and emerin in addition to BAF. It is also necessary for efficient localization of MAN1 to the inner nuclear membrane (*Mansharamani and Wilson, 2005*). The carboxyl-terminal nucleoplasmic region (residues 655–911) exhibits two globular domains (*Pan et al., 2005*; *Caputo et al., 2006*). The first globular domain contains a winged helix (including the sequence RKKMKKVWDR) which is mainly used for DNA binding and recognition. The second domain (amino acids 782–911) is

an RRM-like protein interaction domain where it can interact with R-SMADs. The entire carboxy-terminal region of MAN1 was shown to participate in DNA binding, and this interaction is synergistic to the binding of MAN1 to different transcriptional regulators, including R-SMADs (*Osada et al., 2003*; *Raju et al., 2003*; *Lin et al., 2005*; *Pan et al., 2005*). Consistently, we found that both RRM and DNA binding domains are required for the activation of the *BMAL1* promoter by MAN1. Also congruent with previous findings, we found that mutating three of the highly positively charged and conserved amino acids within the winged helix region is sufficient to dampen the activation efficiency of MAN1. In contrast to previous reports, we found that MAN1 further augments (but not antagonizes) the positive effect of SMAD2 on *BMAL1*. Consistent with this result, we also found that the substitution mutation YV-DD of MAN1 does not influence its effect on *BMAL1* (though RRM domain is required). Collectively, these results suggest that the effects of MAN1 and SMAD2 on *BMAL1* might not be completely independent to each other and that MAN1 might interact with SMAD2 in more than one way (presumably through other protein partners) to modulate transcription of target genes.

Many of our body functions manifest a daily rhythm which is maintained by the rhythmic regulation of approximately 15% of genes by the core molecular clock (*Vollmers et al., 2009*; *Menet et al., 2012*). However, cells must be flexible enough to allow for responses to exogenous and endogenous stimuli. This regulation is likely to be mediated not solely by the molecular clock, but also by many additional global and local mechanisms including at the level of chromatin and genome organization (*Aguilar-Arnal et al., 2013*; *Hubner et al., 2013*). Genetic loci associated with the nuclear lamina through large regions of chromatin (lamin associated domains–LADs) are associated with changes in transcriptional status (*Peric-Hupkes et al., 2010*). Circadian genes or genes involved in rhythmic processes display robust rhythmic expression patterns at the level of nascent mRNA and mRNA (*Menet et al., 2012*; *Rodriguez et al., 2012*), suggesting prominent contribution of transcriptional regulation to clock gene expression. Interestingly, genes harboring this expression pattern are dramatically enriched for specific function in transcriptional regulation and chromatin organization (*Menet et al., 2012*). These unbiased genome-wide transcriptome reports raised the possibility that components of nuclear envelope may modulate oscillation of clock genes at transcriptional levels. A recent study has also demonstrated that the molecular clock drives circadian changes in spatial and temporal chromosomal organization (*Aguilar-Arnal et al., 2013*). Indeed, our study links the nuclear periphery with circadian regulation via the regulatory effects of MAN1 on *BMAL1* through transcription.

This effect is evident not only in mammalian systems, but also in flies, as over-expressing *MAN1* resulted in significantly increased levels of *cyc* mRNA. MAN1 over-expression also increased *tim* mRNA levels, which may be at least partially due to increased *cyc*. Knocking down *MAN1* did not alter *cyc* levels, possibly because the residual MAN1 is still sufficient to maintain normal *cyc* levels. However, this manipulation lengthened behavioral period, suggesting that MAN1 may target other clock components in addition to *cyc*. Over-expressing *MAN1* and *LBR* in flies led to a lengthened behavioral period, while in U2OS cells, these manipulations resulted in moderate shortening of the period. On the other hand, over-expressing and knocking down *Lam* in flies shortened and lengthened the period, respectively, which is consistent with the mammalian data. These discrepancies reflect the differences between mammalian and insect clock, which has been implied in previous work as well (*Lowrey et al., 2000*; *Preuss et al., 2004*; *Xu et al., 2005*). Nevertheless, our results indicate that there is a conserved role for NE components in setting the clock in organisms ranging from invertebrates to humans.

Although only 22–30% of cycling mRNA is driven transcriptionally (*Koike et al., 2012*; *Menet et al., 2012*), demonstration of involvement of the nuclear envelope in regulation of the molecular circadian clock suggests one pathway through which the nuclear envelope may globally and temporally regulate large numbers of genes. Interestingly, expression of the genes for many nuclear envelope proteins also oscillates. This finding sheds new light on the interconnectedness of these biological processes and provides further insight into the mechanism whereby cellular, metabolic, physiological, and behavioral processes that oscillate are modulated in a highly coordinated manner.

## Materials and methods

### Plasmids

To generate FLAG-tagged proteins, human *LBR*, *LMNB1*, and *MAN1* genes were subcloned into pCMV-Tag2A (Stratagene; La Jolla, CA). HA-tagged SMAD1, FLAG-MAN1 (YV-DD), and FLAG-MAN1 (1–759) were gifts from Dr Kunxin Luo (*Pan et al., 2005*). m*Bmal1*-Luc was generated in Dr Satchidananda Panda's

lab and kindly provided by Dr John Hogenesch (*Vollmers et al., 2008*), while the human *BMAL1*-luc construct was a generous gift from Dr Toru Takumi (*Akashi and Takumi, 2005*). Syrian hamster *Bmal1* in pcDNA3.1 vector was provided by Dr David Weaver (*Kume et al., 1999*). pRL-TK was purchased from Promega (Madison, WI). Mutations of expression constructs were introduced by PCR, and all constructs used in this study were verified by sequencing.

## LumiCycle analysis

A stable U2OS-B6 cell line that expresses a destabilized firefly luciferase gene under the control of the m*Bmal1* promoter was obtained from Dr Satchidananda Panda (*Vollmers et al., 2008*). siRNAs targeted to *LBR*, *LMNB1*, *MAN1*, or *SMAD1* (10 nM, Invitrogen; Carlsbad, CA; see *Supplementary file 1A*) were individually transfected into 35-mm culture dishes using Lipofectamine RNAiMAX (Invitrogen). For overexpression of FLAG-tagged constructs, plasmid (2 µg) was distributed into each well along with FuGENE HD (4 µl, Roche; Switzerland). For co-transfection of *MAN1* siRNA and *Bmal1*, Lipofectamine 2000 (Invitrogen) was used. 24 hr after transfection, cells were synchronized with 100 nM dexamethasone in serum-free DMEM containing 10 mM HEPES (pH 7.5) at 37°C for 2 hr. Following synchronization, the media were replaced with phenol red-free DMEM supplemented with 10 mM HEPES and 40 µM Luciferin-EF (Promega). Cells sealed with coverslips were incubated in a 32-channel LumiCycle (Actimetrics; Evanston, IL) to monitor real-time bioluminescence for 5 days. Data were analyzed using Lumicycle Analysis (Actimetrics).

## Fly behavior experiments and analysis

To over-express NE, *cry*GAL4-39 and *cry*GAL4-16 (*Emery et al., 2000*) were crossed to *MAN1*[GS2297] (Kyoto Stock Center; Japan), UAS*Lam* (*Padiath et al., 2006*) and *LBR*[GS2162] (Kyoto Stock Center). To knock down NE, *cry*GAL4-39;UAS*dcr2* and UAS*dcr2*;*cry*GAL4-16 were crossed to UAS*MAN1RNAi* (3167R-1, NIG; Japan), UAS*LamRNAi* (45,635, VDRC; Vienna) and UAS*LBRRNAi* (KK110508, VDRC). For controls, the UAS and GAL4 lines were crossed to *w*[1118] or *yw* strains. Male progenies were assayed for behavior.

Locomotor activity levels of flies were monitored using Trikinetics Activity Monitors (Waltham, MA) for 7 days of 12 hr light-12 hr dark (LD) conditions followed by 7 days of constant darkness (DD). For DD rhythmicity, chi-squared periodogram analyses were performed using Clocklab (Actimetrics). Rhythmic flies were defined as those in which the chi-squared power was ≥10 above the significance line. Period calculations also considered all flies with rhythmic power ≥10.

## Quantitative real-time PCR (qRT-PCR)

RNeasy Mini Kit (Qiagen) was used to isolate total RNAs from synchronized U2OS-B6 cells that were collected at interval of 4 hr over the course of 48 hr. Purified RNA (2 µg) was applied in 20-µl reactions for RT primed with Oligo(dT)$_{20}$ using Super-Script III First-Strand Synthesis System (Invitrogen). All qPCR reactions were carried out on a Rotor-Gene RG-3000 (Corbett Research; Netherland)/or 7900HT Fast Real-Time PCR System (Life technologies; Carlsbad, CA) using FastStart SYBR Green Master (Rox) (Roche). The templates were denatured at 95°C for 10 min, followed by forty cycles with 15 s at 95°C, 10 s at 58°C (Rotor-Gene) or 60 s at 60°C (HT7900 system and data acquisition at the end of this step), or 40 s at 72°C, and an additional 2 s for data acquisition (Rotor-Gene). The standard curve and delta–delta CT methods were used for quantification (Applied Biosystems; Carlsbad, CA). Primers used for expression analysis are listed in *Supplementary file 1B*. Primers used for ChIP assay are labeled as the nucleotide distance from the transcriptional start site (TSS) and +1 indicates the starting of TSS.

Fly heads were isolated at the indicated time points and total RNA was isolated with TRIzol reagent (Invitrogen). After the removal of contaminating genomic DNA by RQ1 DNase (Promega) digestion, total RNA was directly amplified with the QuantiTect SYBR green RT-PCR kit (QIAGEN). The following primers were used: for *cyc*, cyc_110.f 5′-GAGGTCTTCGTCGGAAAGG-3′ and cyc_347.r 5′-AAAGCACAT GGGAATCATGG-3′; for *tim*, tim.f 5′-CTGGGGAGTGACCATGG-3′ and tim.r 5′-GCTGGAATCGCC ACTG-3′; for d*MAN1*, dMAN1_148.f 5′-ATTTTGGCCTGTGACACTGC-3′ and dMAN1_303.r 5′-GAAG CCGCTCTGGATTAGC-3′; for d*Lam*, dLam_446.f 5′-CGAGGGAGCTCAAGAACAAGC-3′ and dLam_675_r. 5′-GCGACAGTGTCTCCTGTTCC-3′; for d*LBR*, dLBR_645.f 5′-CATTGACCACCAACACATCC-3′ and dLBR_825.r 5′-GTTATGCGTTTGCGAATGG-3′; for d*Actin*, dActin.f1 5′-CTAACCTCGCCCTCTCCTCT-3′ and dActin.r1 5′-GCAGCCAAGTGTGAGTGTGT-3′. All other primers used for fly tissues are previously published (*Lim et al., 2007*; *Kilman et al., 2009*).

## Mouse behavior experiment and analyses

*Lmnb1*$^{+/\Delta}$ mice with a targeted disruption of the *Lmnb1* gene (***Vergnes et al., 2004***) or *LMNB1*$^{BAC}$ mouse model overexpressing lamin B1 (***Heng et al., 2013***) was generated as previous described. The animals used here were derived from these mice and have been backcrossed to a C57BL/6J background for at least 10 generations. Wild-type littermates were used in pairs for subsequent experiments. Mice housed in light-tight, sound-attenuated cabinets were entrained to LD cycle for 14 days and then released into DD. Wheel-running activity of mice were monitored using Clocklab (Actimetrics). For DD rhythmicity, chi-squared periodogram analyses were performed using Clocklab. Experiments were approved by the Institutional Animal Care and Use Committee at University of California, San Francisco.

## Western blot

Brains or livers were collected from mice that were entrained in LD cycle for 14 days and were then released into DD. Total cellular proteins were extracted from mouse brain or liver using RIPA buffer (150 mM NaCl, 1% NP-40, 0.1% SDS, 0.5% sodium deoxycholate, 20 mM Tris, pH 7.5, and 5 mM EDTA). Protein lysates from cells were prepared in SDS-PAGE loading buffer. Equal amounts of protein were resolved on 8% SDS-PAGE gels and then transferred to nitrocellulose membrane. After incubation with primary antibody at 4°C overnight, membranes were incubated with secondary antibodies at room temperature for 1 hr. The primary antibodies were anti-LBR rabbit polyclonal antibody (1:500; Abcam), anti-LMNB1 rabbit polyclonal antibody (1:1000; Abcam; England), anti-MAN1 rabbit polyclonal antibody (1:3000; from Dr Kunxin Luo) (***Pan et al., 2005***), anti-mPER2 rabbit polyclonal antibody (1:500; Alpha Diagnostic International; San Antonio, TX), anti-GAPDH mouse monoclonal antibody (1:5000; Chemicon; Billerica, MA), anti-BMAL1 goat polyclonal antibody (1:500; Santa Cruz; Dallas, TX), anti-CLOCK rabbit polyclonal antibody (1:1000; Santa Cruz), and anti-FLAG M2 antibody (1:5000; Sigma; St Louis, MO). The MAN1 antibody was generated by immunizing rabbits with C-terminal peptide (SHLRLRTGLTNSQGSS) of human MAN1 (1:1000; Covance and Agbio, Inc; Princeton, NJ). Secondary antibodies were conjugated either with IRDye 680 or IRDye 800 (LI-COR Biosciences; Lincoln, NE) and visualized with an Odyssey Infrared Imaging System (LI-COR Biosciences).

## Luciferase assays

HEK293 cells were cultured in 24-well plates in DMEM containing 10% fetal bovine serum 24 hr prior to transient transfection with FuGENE HD (Roche) for overexpression (50–200 ng cDNA constructs), or Lipofectamine 2000 (Invitrogen) for siRNA knockdown (8–13 pmol, Invitrogen). All transfection mixtures included a *Renilla* luciferase plasmid (pRL-TK; 0.7 ng), as well as a reporter construct consisting of firefly luciferase driven by mouse *Bmal1* or human *BMAL1* promoter (50 ng). We assayed the *Bmal1/BMAL1* promoter luciferase activity using the Dual-Luciferase Reporter Assay System (Promega), modifying the protocol to use 30 µl of luciferase substrate and Stop-n-Glo/substrate mix for each reaction. The luciferase activity was quantified with a TECAN GENios Pro Microplate Reader (TECAN; Switzerland) 48 hr after the initial transfection. Luciferase reporter vector used is pGL3-basic.

## Chromatin immunoprecipitation (ChIP) assay

We performed ChIP assays using Millipore's EZ-ChIP assay kit (cat. # 17–371; Millipore; Billerica, MA) and protein-G sepharose. In brief, HEK293 cells were transfected with h*BMAL1*-luciferase (3.4 Kb) plus vector, FLAG-tagged WT or truncated DNA-binding constructs of MAN1 as indicated. Cell lysates were sonicated on ice using Branson digital sonifier #250 and 1% of cell lysate was taken as input sample. After incubation with FLAG M2 antibody (Sigma), antibody-loaded protein G agarose beads were washed with cold wash buffer six times followed by low-salt buffer, high salt wash buffer, LiCl wash buffer, and then once with TE (10 mM Tris–HCl at pH 8.0, 1 mM EDTA at pH 8.0). After washing, the beads were re-suspended in 100 µl of ChIP elution buffer supplemented with proteinase K and incubated for 2 hr at 65°C followed by 10 min at 95°C. The beads were spun down and the supernatant was saved. DNA was recovered from the spin column and resuspended in 50 µl of TE, and a 1 µl portion was used for qRT-PCR. The PCR products were analyzed by qRT-PCR and quantitated using 7900HT Fast Real-Time PCR System (Life technologies).

## Statistics

Statistical analyses were performed using the unpaired Student's *t* test, one-way ANOVA with Newman–Keuls test, or two-way ANOVA (Prism5, GraphPad; La Jolla, CA). Data are presented as Mean ± SEM or SD. Significant differences ($p < 0.05$) are marked with asterisks in figures.

## Acknowledgements

We thank Drs S Panda and J Hogenesch for m*Bmal1*-luc construct; Dr T Takumi for h*BMAL1*-luc construct; Dr D Weaver for pcDNA3.1-*Bmal1* construct; and Dr K Luo for MAN1 constructs, HA-SMAD1 construct, and MAN1 antibody.

## Additional information

### Competing interests

LP: Reviewing editor, *eLife*. The other authors declare that no competing interests exist.

### Funding

| Funder | Grant reference number | Author |
|---|---|---|
| National Institutes of Health | HL059596 | Louis Ptáček, Ying-Hui Fu |
| National Institutes of Health | GM079180 | Louis Ptáček, Ying-Hui Fu |
| National Institutes of Health | NS072360 | Louis Ptáček, Ying-Hui Fu |
| Howard Hughes Medical Institute | | Louis Ptáček |

The funders had no role in study design, data collection and interpretation, or the decision to submit the work for publication.

### Author contributions

S-TL, LZ, Conception and design, Acquisition of data, Analysis and interpretation of data, Drafting or revising the article, Contributed unpublished essential data or reagents; XL, Conception and design, Acquisition of data, Analysis and interpretation of data, Contributed unpublished essential data or reagents; LCZ, Acquisition of data, Analysis and interpretation of data, Drafting or revising the article; VEG, C-WT, Acquisition of data, Analysis and interpretation of data; LP, Y-HF, Conception and design, Analysis and interpretation of data, Drafting or revising the article, Contributed unpublished essential data or reagents

### Ethics

Animal experimentation: All experiments were performed in accordance with the National Institutes of Health Guide for the Care and Use of Laboratory Animals and approved by Institutional Animal Care and Use Committee of University of California San Francisco. The protocol was approved by the Committee on the Ethics of Animal Experiments of the University of California San Francisco (AN089663-01).

## Additional files

### Supplementary file

• Supplementary file 1. (**A**) Sequences of siRNAs used in the study. (**B**) qRT-PCR primer sequences for human clock genes.

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
