## [Decision Letter]

Thank you for sending your work entitled “Nuclear envelope protein MAN1 regulates clock through *BMAL1*” for consideration at *eLife*. Your article has been favorably evaluated by a Senior editor and 3 reviewers, one of whom is a member of our Board of Reviewing Editors.

The following individuals responsible for the peer review of your submission have agreed to reveal their identity: Leslie Griffith, BRE; Satchidananda Panda, reviewer.

The Reviewing editor and the other reviewers discussed their comments before we reached this decision, and the Reviewing editor has assembled the following comments to help you prepare a revised submission.

This paper provides a link between the nuclear envelope and the circadian clock and posits that the link is evolutionarily conserved, by suggesting that regulation of the clock by NE proteins, in particular MAN1, is present in both rodents and flies. The novelty of the study is high and the data are mostly clear and compelling. There are, however, a few weak links in the story which, if strengthened would make it quite compelling. The major issues are:

1) At the organismal/cellular level: The fly data generally are not completely convincing. It’s very easy to get RNAi phenotypes and a lot of times these don't map to the gene targeted. Phenotypes based on single RNAis, with no mutant data or experiments mapping the effect to the relevant gene, are not very convincing. In addition, the data in Table 1 for human LMB1 do not support what is stated in the text- there appears to be no change in period. Some genetic evidence that the RNAi phenotypes are actually due to changes in the relevant genes is required.

2) At the molecular mechanism level: The cell assay data are quite subtle. If the authors could show suppression of the long period of the NE knockdown cells (or knockout mouse fibroblasts) this would provide very strong evidence of the linkage the authors wish to make.

Reviewer #1:

This is a very interesting and novel study which suggests that NE proteins can directly modulate the circadian clock in both mammals and insects. Gene regulation by the nuclear envelope is a very new concept and connecting it to the clock is a potentially very important observation.

For the most part the data are very clear and consistent. The only piece of evidence that is less than clear is in Figure 1. How do the authors draw the conclusion that MAN1 is cycling? It looks dead flat to me. Is there any evidence to suggest that the very small difference statistically significant? Ditto for LMNB1 and LBR. They are by eye a bit more convincing, but a quantitative treatment is required if they authors want to claim the protein cycles (which I am not sure they need to given the strength of the functional evidence- so what if the protein does not cycle in an extract of liver?).

Reviewer #2:

The manuscript demonstrates that nuclear envelope proteins contribute to the determination of circadian period. It focuses in particular on the MAN1 protein, showing that this is a positive regulator of BMAl1 transcription. Knockdown or overexpression of MAN1 or other nuclear envelope (NE) proteins, Lamin B1 and LBR, produces a small effect on circadian period in mammalian cultured cells. A more robust phenotype is seen with some of these manipulations in *Drosophila* although the directionality of the effect (short or long period) or the mechanism is not easily explained. Microarray studies have reported cyclic expression of nucleoporins, and not surprisingly, the authors report cyclic expression for a couple of other NE proteins. Overall, these are interesting observations although they do not quite provide a picture of how the nuclear envelope Figure in the clock mechanism. Additional comments are below:

Figure 1 should show quantification of multiple blots for PER2 cycling in the LaminB1 mutant and overexpression backgrounds (Figure 1). Also, what happens to other clock proteins in these backgrounds?

Effects of knockdown and over-expression of NE proteins in mammalian cells are subtle. In mice heterozygous for a Lamin B1 mutation, or overexpressing LaminB1, PER2 oscillations appear to be altered, although this needs to be better documented, and is not linked to any overt rhythm. Period lengthening by overexpression of human MAN1 or LBR in *Drosophila* has a more robust effect, but then knockdown of MAN1 has the same effect. Given that the molecular mechanism is not known (see below for the inconclusive data regarding the effect on cyc), and mutant data are not shown (only a single RNAi for each gene, without any mapping of the RNAi effect) it is difficult to assess the significance of the fly data.

The text indicates that over-expression of human LMNB1 shortens the period in *Drosophila*, but this is not supported by the data in Table 1.

Given that knocking down MAN1 has no effect on cyc, the effect of overexpression could be non-physiological. Also, increased expression of cyc (through MAN1 overexpression) should not lengthen period as shown here, so it is unlikely that the effect on circadian period is going through cyc. Along the same lines, it would be good to know if the effects on circadian period in mammalian cells are going through BMal1. Does overexpression of BMal1 rescue the long period produced by knockdown of NE proteins?

Additive effects of SMAD and MAN1 suggest that they are acting independently, and not as part of the same pathway.

The studies mapping MAN1 interactions with the BMAL1 promoter are good, and indicate direct interaction.

Reviewer #3:

This manuscript offers a novel link between nuclear envelope components and circadian clock that is conserved from insects to mammals. The experiments are straightforward and well described. However, some gaps in data presentation need to be addressed.

Since the authors have access to the lamin mutant mice and flies with specific perturbation of Ne components, figure (a) showing the circadian behavior phenotype of these animals will substantially strengthen the manuscript. If the het mice lack any appreciable defect in free running circadian period, activity consolidation, phase angle of entrainment, such lack of phenotype may be explained in the discussion section. Circadian behavior phenotype of the flies should be shown in the main figures to strengthen the manuscript.

siRNA knockdown experiments on Bmal1:Luc expression should ideally have another promoter:Luc construct to rule out the possibility that the down regulation of Bmal1:luc signal is not due to general defect in nuclear integrity or overall transcription.

---

## [Author Response]

*1) At the organismal/cellular level: The fly data generally are not completely convincing. It’s very easy to get RNAi phenotypes and a lot of times these don't map to the gene targeted. Phenotypes based on single RNAis, with no mutant data or experiments mapping the effect to the relevant gene, are not very convincing. In addition, the data in*
Table 1
*for human LMB1 do not support what is stated in the text- there appears to be no change in period. Some genetic evidence that the RNAi phenotypes are actually due to changes in the relevant genes is required*.

In our original submission, we showed that MAN1 mRNA level is reduced in flies expressing MAN1-RNAi. Here we have included additional data demonstrating that Lam and LBR mRNA levels are reduced in flies expressing Lam-RNAi and LBR-RNAi, respectively (Figure 2—figure supplement 2). This does not eliminate the possibility that the RNAi is also disrupting expression of other genes. However, given that we also observe circadian phenotypes when over-expressing these genes in flies, and that deficiency of these genes lead to disrupted circadian clock in the mammalian system, we believe it is highly likely that the circadian phenotypes associated with expressing RNAi constructs of MAN1/Lam/LBR in flies are due to reduction in the levels of these genes.

As for Lam over-expressing flies, the behavioral period is shorter than the GAL4 controls, which have long periods, but not shorter than UASLam control, which has a ∼24hr period. It is possible that there is a limit to how much Lam over-expression can shorten the period in flies. It may be able to shorten period to ∼24hr but cannot further shorten period to less than 24hr.

*2) At the molecular mechanism level: The cell assay data are quite subtle. If the authors could show suppression of the long period of the NE knockdown cells (or knockout mouse fibroblasts) this would provide very strong evidence of the linkage the authors wish to make*.

Thank you for the suggestion. We have added new data demonstrating that Bmal1 over-expression suppresses the effect of MAN1 knockdown on circadian period in U2OS cells (Figure 4—figure supplement 2).

Reviewer #1:

*This is a very interesting and novel study which suggests that NE proteins* can *directly modulate the circadian clock in both mammals and insects. Gene regulation by the nuclear envelope is a very new concept and connecting it to the clock is a potentially very important observation*.

*For the most part the data are very clear and consistent. The only piece of evidence that is less than clear is in*
Figure 1*. How do the authors draw the conclusion that MAN1 is cycling? It looks dead flat to me. Is there any evidence to suggest that the very small difference statistically significant? Ditto for LMNB1 and LBR. They are by eye a bit more convincing, but a quantitative treatment is required if they authors want to claim the protein cycles (which I am not sure they need to given the strength of the functional evidence- so what if the protein does not cycle in an extract of liver?)*.

Quantification of the Western blot is shown in Figure 1 top panel. The reviewer is correct that the oscillation for MAN1 is mild. However, the subsequent functional data strengthened our original idea that MAN1 could be the link between NE and the molecular clock.

Reviewer #2:

*The manuscript demonstrates that nuclear envelope proteins contribute to the determination of circadian period. It focuses in particular on the MAN1 protein, showing that this is a positive regulator of BMAl1 transcription. Knockdown or overexpression of MAN1 or other nuclear envelope (NE) proteins, Lamin* B1 *and LBR, produces a small effect on circadian period in mammalian cultured cells. A more robust phenotype is seen with some of these manipulations in Drosophila although the directionality of the effect (short or long period) or the mechanism is not easily explained. Microarray studies have reported cyclic expression of nucleoporins, and not surprisingly, the authors report cyclic expression for a couple of other NE proteins. Overall, these are interesting observations although they do not quite provide a picture of how the nuclear envelope Figure in the clock mechanism. Additional comments are below:*

Figure 1
*should show quantification of multiple blots for PER2 cycling in the LaminB1 mutant and overexpression backgrounds (*Figure 1*). Also, what happens to other clock proteins in these backgrounds?*

We have included quantification of PER2 Western in Figure 1. We did not examine other clock proteins at the time of experimentation and thought that PER2 was the most logical representative clock protein for us to test. Unfortunately, the tissue samples were lost in a freezer breakdown in the lab and the mouse lines have been cryopreserved. Although we would like to examine the other clock proteins as well, it would take tremendous amount of time and effort to re-derive these animals and perform the experiments

*Effects of knockdown and over-expression of NE proteins in mammalian cells are subtle. In mice heterozygous for a Lamin* B1 *mutation, or overexpressing LaminB1, PER2 oscillations appear to be altered, although this needs to be better documented, and is not linked to any overt rhythm*.

From the Western blot in Figure 1, the peak PER2 oscillation time for heterozygous Lmnb1 is CT 0 vs CT 20 for WT, and the peak time for LMNB^BAC^ is clearly CT 16. Either way, there is a 4 hr phase shift compared to control. Although we did not show the same shift in wheel-running rhythm, we reason that this may be due to the resilience of the SCN clock to perturbations. Therefore we can only observe alterations in peripheral clocks. It is also possible that other molecular changes in addition to PER2 can somehow neutralize the behavior rhythm. We have added this discussion in the text.

*Period lengthening by overexpression of human MAN1 or LBR in Drosophila has a more robust effect, but then knockdown of MAN1 has the same effect. Given that the molecular mechanism is not known (see below for the inconclusive data regarding the effect on cyc), and mutant data are not shown (only a single RNAi for each gene, without any mapping of the RNAi effect) it is difficult to assess the significance of the fly data*.

Please see point 1 (General) above. As addressed earlier, we have added additional data validating that Lam and LBR are knocked down by RNAi. It is true that not all changes are consistent between mammals and fly, and there is still need for future investigation to understand the differences between these two systems. We have added discussion in the text regarding this point.

*The text indicates that over-expression of human LMNB1 shortens the period in Drosophila, but this is not supported by the data in*
Table 1.

We apologize for not making this clear in the original submission. When comparing cryGAL-39/UASLam to cryGAL4-39/+, or UASLam/+;cryGAL4-16/+ to cryGAL4-16/+, there are shortened periods. We have modified the text to clarify this.

*Given that knocking down MAN1 has no effect on cyc, the effect of overexpression could be non-physiological. Also, increased expression of cyc (through MAN1 overexpression) should not lengthen period as shown here, so it is unlikely that the effect on circadian period is going through cyc*.

We agree that over-expression of MAN1 in flies drives MAN1 much higher than the endogenous level, which may be “non-physiological”, but regardless, it demonstrates in this in vivo system, that increasing MAN1 results in increased cyc expression, consistent with what we demonstrate in cell culture, i.e. increasing MAN1 leads to increased Bmal1 expression. We were not able to demonstrate that knocking down MAN1 reduces cyc in vivo as we showed for Bmal1 in vitro, but we believe this is probably because in vivo knockdown is not as efficient as in vitro, and/or that the in vivo system is more resilient to perturbations compared to in vitro. Thus we do believe that Bmal1/cyc is a target of MAN1 in the clock, although it may not be the only target. When MAN1 is over-expressed in flies, the period is lengthened and cyc appears to be the clock gene that exhibits the largest changes, leading us to suspect that under the condition of MAN1 over-expression, cyc is the primary target of MAN1 in the clock. However, it is certainly possible that MAN1 is also targeting other clock genes. When MAN1 is knocked down in flies, cyc is not affected while period is still lengthened, suggesting that under MAN1 deficient condition, some other clock genes are altered which contribute to the period phenotype. Therefore, we can only draw the conclusion that in MAN1 knock-down flies, the period phenotype is not going through cyc, but we cannot draw the same conclusion for MAN1 over-expressing flies.

Along the same lines, it would be good to know if the effects on circadian period in mammalian cells are going through BMal1. Does overexpression of BMal1 rescue the long period produced by knockdown of NE proteins?

Thank you for the suggestion. As mentioned above in point 2 (General), we have included new data to address this question (Figure 4—figure supplement 2).

*Additive effects of SMAD and MAN1 suggest that they are acting independently, and not as part of the same pathway*.

We agree with the reviewer that this would seem to be the case at first. However, since MAN1’s enhancing effect on BMAL1 requires its RRM domain which is the SMAD interacting domain and the only SMAD that has effect on BMAL1 is SMAD2. For these reasons, MAN1 and SMAD2 may not be acting completely independently.

*The studies mapping MAN1 interactions with the BMAL1 promoter are good, and indicate direct interaction*.

Reviewer #3:

*This manuscript offers a novel link between nuclear envelope components and circadian clock that is conserved from insects to mammals. The experiments are straightforward and well described. However, some gaps in data presentation need to be addressed*.

*Since the authors have access to the lamin mutant mice and flies with specific perturbation of Ne components, figure (a) showing the circadian behavior phenotype of these animals will substantially strengthen the manuscript. If the het mice lack any appreciable defect in free running circadian period, activity consolidation, phase angle of entrainment, such lack of phenotype may be explained in the discussion section*.

Neither Lamin b1 heterozygous mutant nor Lamin B1 over-expressing mice show altered wheel-running rhythms (Figure 1—figure supplement 1). We have included these data along with relevant discussion in the revised manuscript.

*Circadian behavior phenotype of the flies should be shown in the main figures to strengthen the manuscript*.

We have added a new figure to demonstrate the behavioral profiles of flies with NE over-expressed or knocked-down (Figure 2—figure supplement 1).

*siRNA knockdown experiments on Bmal1:Luc expression should ideally have another promoter:Luc construct to rule out the possibility that the down regulation of Bmal1:luc signal is not due to general defect in nuclear integrity or overall transcription*.

For in vitro transcription assay, Renilla-luc was always used as internal control. Since Renilla-luc was not affected by siRNA, we believe that the effect we observed for Bmal-luc is not due to the general defect in nuclear integrity or overall transcription. In addition, the new data of Bmal1 overexpression rescuing the period change in U2OS cells further demonstrates that the down regulation of Bmal1:luc signal is not due to general defect in nuclear integrity or overall transcription.